# AGDC: Autoregressive Generation of Variable-Length Sequences with Joint Discrete and Continuous Spaces

## Abstract

Transformer-based autoregressive models excel in data generation but are inherently constrained by their reliance on discretized tokens, which limits their ability to represent continuous values with high precision. We analyze the scalability limitations of existing discretization-based approaches for generating hybrid discrete-continuous sequences, particularly in high-precision domains such as semiconductor circuit designs, where precision loss can lead to functional failure. To address the challenge, we propose **AGDC**, a novel unified framework that *jointly models discrete and continuous values for variable-length sequences*. AGDC employs a hybrid approach that combines categorical prediction for discrete values with diffusion-based modeling for continuous values, incorporating two key technical components: an end-of-sequence (EOS) logit adjustment mechanism that uses an MLP to dynamically adjust EOS token logits based on sequence context, and a length regularization term integrated into the loss function. Additionally, we present **ContLayNet**, a large-scale benchmark comprising 334K high-precision semiconductor layout samples with specialized evaluation metrics that capture functional correctness where precision errors significantly impact performance. Experiments on semiconductor layouts (ContLayNet), graphic layouts, and SVGs demonstrate AGDC's superior performance in generating high-fidelity hybrid vector representations compared to discretization-based and fixed-schema baselines, achieving scalable high-precision generation across diverse domains.

## 1 Introduction

Autoregressive models (Vaswani et al., 2017; Brown et al., 2020; Ramesh et al., 2021; Radford et al., 2023) have shown remarkable success in generating sequences across various domains, typically relying on discrete values such as text tokens and quantized image representations. However, many real-world applications inherently require hybrid representations that combine discrete and continuous values in variable-length sequences. Layout designs integrate discrete component types with precise continuous positions, Scalable Vector Graphics (SVG) pair discrete drawing commands with continuous coordinates, and music notation combines discrete notes with continuous timing. Each of these systems naturally varies in sequence length depending on content complexity.

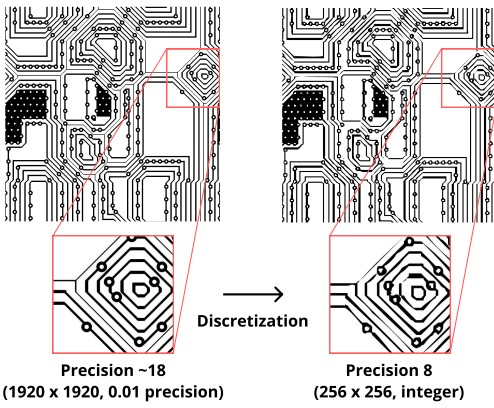

**Precision ~18**
**(1920 x 1920, 0.01 precision)**

Discretization

**Precision 8**
**(256 x 256, integer)**

Figure 1: **Impact of discretization.** Discretization fundamentally compromises the scalability of precision. For precision analysis, refer to Section 3.1.

Conventional approaches to handling hybrid discrete-continuous sequences face fundamental limitations. These methods typically discretize continuous values into discrete tokens (Gupta et al., 2021; Wu et al., 2023; 2021; Gulati & Roysdon, 2023) or rely on image rasterization (Frans et al., 2022; Jain et al., 2023; Xing et al., 2024), both fundamentally compromising precision by imposing

Table 1: **Comparison with existing works.** AGDC is the first autoregressive framework to jointly model hybrid discrete-continuous representations while supporting variable-length sequence generation.

| | Hybrid? | Autoregressive? | Variable-Length? | Domain |
|---|---|---|---|---|
| LT (Gupta et al., 2021) | | ✓ | ✓ | Layout |
| IconShop (Wu et al., 2023) | | ✓ | ✓ | SVG |
| DLT (Levi et al., 2023) | ✓ | | △ | Layout |
| DP-TBART (Castellon et al., 2023) | | ✓ | | Tabular Data |
| TabNAT (Zhang et al., 2025) | ✓ | | | Tabular Data |
| **AGDC (Ours)** | **✓** | **✓** | **✓** | **Domain-Agnostic** |

finite resolution limits (Figure 1). This precision sacrifice becomes particularly critical in high-precision domains such as semiconductor circuit design, where even minor positioning errors can cause complete system failure. Recently, MAR (Li et al., 2024) extended autoregressive modeling to purely continuous sequences for image generation, but remains limited to fixed-sized image outputs. Meanwhile, few existing methods (Levi et al., 2023; Zhang et al., 2025) successfully combine discrete and continuous values by preserving their nature, but are limited to fixed schemas (tabular data) or rely on non-autoregressive diffusion, both of which are inadequate for variable-length sequential generation.

To address these challenges, we propose **AGDC**, a novel autoregressive framework that jointly models discrete and continuous values while supporting variable-length generation (Table 1). Specifically, discrete values are predicted through traditional categorical prediction, while continuous values are handled via diffusion-based probabilistic models. Our method provides a unified latent representation of both value types, seamlessly integrating them within an autoregressive structure. To generate outputs of contextually appropriate lengths, we propose two technical components. First, we introduce an end-of-sequence (EOS) logit adjustment mechanism that employs an MLP to dynamically adjust EOS token logits based on the context. Second, we incorporate a length regularization term into the loss function, enabling differentiable length control during training. We demonstrate through our experiments that these components help the model generate sequences closer to ground-truth lengths, thus achieving performance enhancement.

In addition, we introduce **ContLayNet**, a benchmark specifically designed to evaluate hybrid discrete-continuous sequence generation in real-world engineering applications. The ContLayNet dataset comprises 334K nano-scale semiconductor layout samples with naturally variable-length sequences, represented as high-precision vectors and collected from real-world sources. This benchmark addresses the scarcity of datasets, where precision errors significantly impact functional performance, enabling rigorous evaluation of hybrid generation methods. To facilitate a comprehensive evaluation, we propose specialized metrics based on Design Rule Checks (DRC).

The main contributions of this paper are summarized below:

- We reveal the core limitations of existing deep learning methods in handling hybrid discrete-continuous vector data, especially those relying on discretization, which often compromise precision or structural integrity.

- We propose AGDC, a novel autoregressive framework that jointly models discrete and continuous values within variable-length sequences, effectively bridging discrete and continuous components without lossy transformations. To make output sequences contextually appropriate in length, we introduce an MLP-based EOS logit adjustment mechanism and a length regularization term that enables differentiable length control during training.

- We construct ContLayNet, a large-scale benchmark comprising high-precision hybrid vector representations of real-world semiconductor layouts, alongside specialized Design Rule Checks (DRC) metrics to enable rigorous evaluation of generative models in this domain.

- We demonstrate AGDC's effectiveness across multiple domains—semiconductor layouts, graphic layouts, and text-to-SVG synthesis—achieving superior performance in high-precision settings while generating variable-length hybrid sequences, outperforming existing discretization-based and fixed-schema methods.

## 2 RELATED WORKS

Current machine learning approaches for vector representation employ various techniques, but most compromise precision. Discretization methods convert continuous coordinates into discrete tokens (Gupta et al., 2021; Wu et al., 2023; 2021), while rasterization-based approaches (Jain et al., 2023; Xing et al., 2024) employ differentiable renderers (Li et al., 2020) or leverage pretrained vision models (Frans et al., 2022; Xing et al., 2023) like CLIP (Radford et al., 2021)—fundamentally limiting vector optimization by tying it to raster representation constraints. Despite impressive results, these methods face inherent limitations in scalability. Only a few approaches (Cao et al., 2023; Levi et al., 2023; Zhang et al., 2025) respect the continuous nature of hybrid vector representations, but are domain-specific or constrained by fixed schemas, limiting their general applicability to complex, variable-length sequences. Recent efforts (Chen et al., 2024; Zhao et al., 2024; Li et al., 2025) have explored combining autoregressive and diffusion models, yet remain limited to specific domains (graph generation with discrete variables) or data types (continuous-only sequences with fixed-length chunks), without addressing the joint modeling of hybrid discrete-continuous representations with variable-length control. Our proposed AGDC, as a domain-agnostic approach, effectively addresses variable-length hybrid vector-represented sequences through a unified autoregressive approach that jointly handles both discrete and continuous values.

## 3 PRELIMINARIES

This section establishes the theoretical foundations of our work. We define precision formulations and highlight scaling challenges in Section 3.1, and then review autoregressive models from traditional discrete approaches to recent continuous extensions, along with their limitations in Section 3.2.

### 3.1 PRECISION ANALYSIS

For each continuous dimension $x \in X$, we define the precision of this dimension as follows:

$$P_x = \log_2 \left( \frac{x_{\max}}{\Delta x} \right) \quad (1)$$

where $x_{\max} = \sup(X)$ is the supremum of the domain, and $\Delta x = \inf(|x_a - x_b|)$ with $x_a, x_b \in X$ and $a \neq b$ is the minimum non-zero distance between distinct points.

Hybrid vector representations preserve continuous values, theoretically supporting *infinite* precision (as $\Delta x \to 0$). This enables exact positioning and scale invariance, which is critical for applications such as semiconductor circuit design, where nanometer-level precision is essential for functional integrity.

In contrast, discretization imposes a lower bound on $\Delta x$. For example, a $200 \times 200$ discrete grid with integer quantization ($\Delta x = 1$) yields only $P_x = P_y \approx 7.6$ bits of precision. To match the precision of continuous representations, the discretized vocabulary must grow exponentially; achieving $P_x$ bits of resolution requires $2^{P_x}$ unique tokens.

This exponential growth creates a critical trade-off between precision and computation; higher precision demands exponentially more tokens, leading to prohibitive computational costs and training instability due to vocabulary explosion. We empirically demonstrate this limitation in the appendix, where we show how increasing the precision in LayoutTransformer (Gupta et al., 2021) leads to performance degradation.

### 3.2 AUTOREGRESSIVE MODELS

Traditional autoregressive models (Vaswani et al., 2017; Devlin et al., 2019; Brown et al., 2020) operate on discrete token spaces, modeling sequences through conditional probabilities:

$$p(x^1, \cdots, x^n) = \prod_{i=1}^{n} p(x^i | x^1, \cdots, x^{i-1}) \quad (2)$$

These models predict discrete tokens from a fixed vocabulary using categorical distributions, creating the precision-computation trade-offs discussed in Section 3.1. Recently, MAR (Li et al., 2024) has

extended autoregressive modeling to continuous spaces using diffusion processes. An autoregressive network produces a conditioning vector $\mathbf{z} \in \mathbb{R}^D$ at each step, which guides a small diffusion network to model the continuous probability distribution $p(x|\mathbf{z})$. However, MAR is designed for fixed-size image generation with purely continuous values. Our AGDC extends beyond this limitation by jointly modeling discrete and continuous components within variable-length sequences, addressing the broader challenge of hybrid vector representations across diverse domains.

## 4 AGDC MODEL

This section presents our autoregressive generation framework designed to process hybrid vector representation datasets. We introduce our atomic unit representation approach for handling hybrid vectors in Section 4.1, followed by our AGDC methodology in Section 4.2.

### 4.1 ATOMIC UNIT REPRESENTATION

Many real-world domains, including layouts and SVGs, naturally contain both discrete (*e.g.*, types, classes) and continuous values (*e.g.*, coordinates, amounts). The majority of such hybrid vector representations can be expressed as sequences of *atomic units*, with each unit combining discrete and continuous components (Figure 2). This paper focuses on these representable domains.

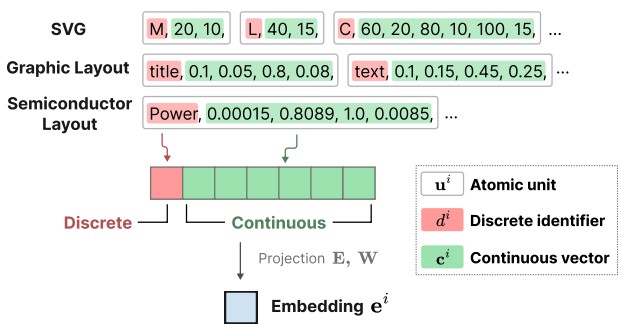

Figure 2: **Atomic unit representation.**

To formalize this representation, consider a sequence

$$\mathbf{U} = [\mathbf{u}^1, \mathbf{u}^2, \cdots, \mathbf{u}^n] \tag{3}$$

composed of atomic units denoted by $\mathbf{u}^i$. Each atomic unit consists of a discrete identifier $d^i$ and a continuous vector $\mathbf{c}^i$ as

$$\mathbf{u}^i = [d^i, \mathbf{c}^i], \tag{4}$$

where $d^i$ takes discrete values in $\{0, \ldots, K-1\}$ and $\mathbf{c}^i = (c_1^i, c_2^i, ..., c_m^i)$ consists of real-valued scalars, $c_j^i$ $(j = 1, \ldots, m)$.

For the example of semiconductor and graphic layouts, an atomic unit is structured as

$$\mathbf{u}^i = [d^i, x^i, y^i, w^i, h^i] \tag{5}$$

where $d^i$ specifies the element type—*e.g.*, power, wiring, and device layers in ContLayNet; text, title, list, table, or figure in PubLayNet (Zhong et al., 2019)—and $(x^i, y^i, w^i, h^i)$ define the position and size, respectively. Similarly, in SVGs, an atomic unit takes the form

$$\mathbf{u}^i = [d^i, x_1^i, y_1^i, \cdots, x_4^i, y_4^i] \tag{6}$$

where $d^i$ denotes the command type (M, L, C) and $(x^i, y^i)$ defines the coordinates of control point.

### 4.2 METHODOLOGY

AGDC leverages a unified architecture that processes discrete and continuous components in parallel, enabling seamless integration of both data types.

**Input embedding.** For each atomic unit $\mathbf{u}^i$ in the input sequence $\{\mathbf{u}^i\}_{i=1}^n$, as defined in Equation (4), we compute its corresponding embedding vector $\mathbf{e}^i \in \mathbb{R}^D$ through:

$$\mathbf{e}^i = \text{concat}(\mathbf{E}(\text{onehot}(d^i)), \mathbf{W}\mathbf{c}^i) \tag{7}$$

where $\mathbf{E} \in \mathbb{R}^{D \times K}$ is a learned embedding matrix that maps discrete tokens to a $D$-dimensional space, and $\mathbf{W} \in \mathbb{R}^{D \times m}$ is a projection matrix that transforms continuous vectors into the same space. The embedding vector $\mathbf{e}^i$ incorporates information from both discrete and continuous components.

**Model architecture.** The autoregressive network $f(\cdot)$ processes the sequence of embedding vectors $(\mathbf{e}^1, \mathbf{e}^2, \cdots, \mathbf{e}^{i-1})$ to generate a latent representation $\mathbf{z}^i \in \mathbb{R}^D$ for the current step $i$:

$$\mathbf{z}^i = f(\mathbf{e}^1, \mathbf{e}^2, \cdots, \mathbf{e}^{i-1}). \tag{8}$$

This latent representation $\mathbf{z}^i$ conditions two parallel branches for discrete and continuous generations. In the discrete branch, $\mathbf{z}^i$ is converted into categorical probabilities using a multi-layer perceptron (MLP) followed by a softmax function:

$$p(d^i|\mathbf{z}^i) = \text{softmax}(\text{MLP}_{\text{DISC}}(\mathbf{z}^i)) \quad (9)$$

In the continuous branch, a denoising network $\varepsilon_\theta$, conditioned on $\mathbf{z}^i$, iteratively removes noise from corrupted target vectors, enabling sampling from the continuous distribution $p(\mathbf{c}^i|\mathbf{z}^i)$. Following MAR (Li et al., 2024), $\varepsilon_\theta$ is implemented as a small MLP consisting of residual blocks with AdaLN (Peebles & Xie, 2023).

**EOS logit adjustment.** This mechanism aims to provide an additional context-dependent signal for appropriate sequence termination and is implemented as

$$\text{logits}_d^i[\text{EOS}] \leftarrow \text{logits}_d^i[\text{EOS}] + \alpha \cdot \text{MLP}_{\text{EOS}}(\mathbf{z}^i),$$

where $\alpha$ is a scaling factor and $\text{MLP}_{\text{EOS}}(\cdot)$ is the logit-adjustment term based on the current sequence state.

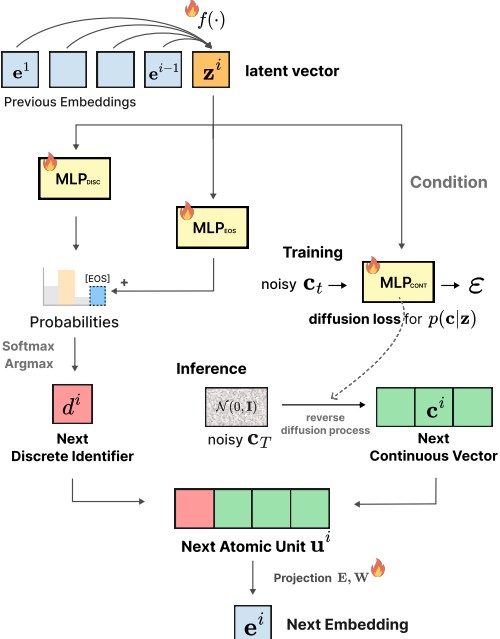

Figure 3: **Overview of AGDC.**

**Training.** The discrete component is trained using Cross-Entropy (CE) loss:

$$\mathcal{L}_d(\mathbf{z}^i, d^i) = \text{CE}(\text{onehot}(d^i), p(d^i|\mathbf{z}^i)) \quad (10)$$

The continuous component is trained using a denoising score-matching objective:

$$\mathcal{L}_c(\mathbf{z}^i, \mathbf{c}^i) = \mathbb{E}_{\varepsilon, t}\left[\|\varepsilon - \varepsilon_\theta(\mathbf{c}_t^i|t, \mathbf{z}^i)\|^2\right] \quad (11)$$

where $\varepsilon \sim \mathcal{N}(\mathbf{0}, \mathbf{I})$ is a Gaussian noise vector and $\mathbf{c}_t^i = \sqrt{\bar{\alpha}_t}\mathbf{c}^i + \sqrt{1 - \bar{\alpha}_t}\varepsilon$ is a noise-corrupted vector at timestep $t$.

We also integrate a *length regularization loss* that encourages generated sequences to align with target lengths during training. After applying the EOS logit adjustment, the expected sequence length is computed from the adjusted EOS probability, which is given by

$$E[\text{length}] = \sum_{t=1}^{T} t \cdot p_{\text{EOS}}^t \prod_{i=1}^{t-1}[1 - p_{\text{EOS}}^i] \quad (12)$$

where $p_{\text{EOS}}^i$ is the probability of the EOS token at position $i$. The length regularization term $\mathcal{L}_\ell$ is then defined by

$$\mathcal{L}_\ell = (E[\text{length}] - L_{\text{target}})^2. \quad (13)$$

This formulation is fully differentiable, enabling end-to-end optimization of sequence lengths alongside content generation. The total training objective combines the two loss components with the regularization term, which is given by

$$\mathcal{L}_{\text{total}} = \mathcal{L}_d + \lambda_1 \cdot \mathcal{L}_c + \lambda_2 \cdot \mathcal{L}_\ell \quad (14)$$

where $\lambda_1$ and $\lambda_2$ balance the three loss terms.

**Generation process.** For discrete identifiers, samples are drawn from $p(d|\mathbf{z})$ after the EOS adjustment. For continuous vectors, samples are generated through a reverse diffusion process:

$$\mathbf{c}_{t-1} = \frac{1}{\sqrt{\alpha_t}}\left(\mathbf{c}_t - \frac{1 - \alpha_t}{\sqrt{1 - \bar{\alpha}_t}}\varepsilon_\theta(\mathbf{c}_t|t, \mathbf{z})\right) + \sigma_t\delta \quad (15)$$

where $\delta \sim \mathcal{N}(\mathbf{0}, \mathbf{I})$ is a noise vector, and $\sigma_t$ controls the noise level. Starting from an initial noise vector $\mathbf{c}_T \sim \mathcal{N}(\mathbf{0}, \mathbf{I})$, this iterative process generates samples that follow the target distribution $p(\mathbf{c}|\mathbf{z})$. By generating each value in its natural form, our approach preserves high precision while maintaining the inherent structure of hybrid vector representations.

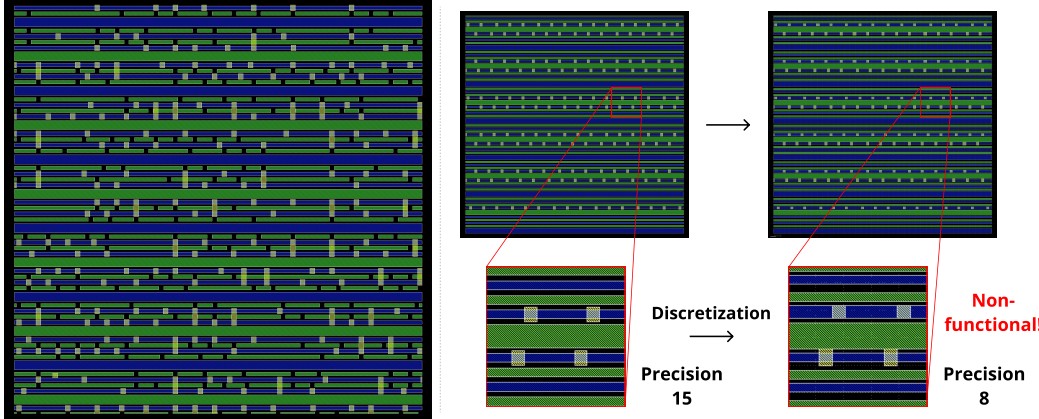

Figure 4: **The ContLayNet dataset.** Left: Example visualization of a sample. Right: Comparison showing functional failure when ContLayNet samples get discretized to low precision.

## 5 CONTLAYNET BENCHMARK

This section provides details on the dataset and evaluation metrics in the ContLayNet benchmark.

### 5.1 DATASET

The ContLayNet dataset consists of 334,330 semiconductor layout samples, with original continuous values converted to an effective 15-bit precision ($40,000 \times 40,000$ integer grid)[1]. Each sample contains three primary layers—Power, Wiring, and Device—visualized in blue, green, and yellow, respectively, in Figure 4. The cells specify each layer's type, location, and size. Power (written as Layer 515), Wiring (written as Layer 644), and Device (written as Layer 1457) layers contain an average of 30, 116, and 178 elements, respectively, with standard deviations of 10, 92, and 75. The dataset presents substantial structural complexity, with each sample averaging 323 layers. Original (continuous) values are divided and rounded-off to a $40,000 \times 40,000$ integer grid with substantial structural complexity, each sample averaging 323 layers.

Figure 4 illustrates how reducing precision leads to functional failures: low-precision discretization disrupts critical geometric alignments between layers, ultimately rendering circuits non-functional. This property makes ContLayNet particularly well-suited to our research objectives—it offers a real-world dataset in which precision loss directly results in measurable performance degradation.

### 5.2 EVALUATION METRICS

Evaluating semiconductor layouts presents unique challenges, as traditional image-based metrics like FID (Heusel et al., 2017) and CLIP Score (Radford et al., 2021) fail to capture the critical internal semantics needed for quality assessment. To address this gap, we introduce specialized evaluation metrics based on Design Rule Checks (DRC), which assess whether generated layouts satisfy essential constraints of real-world circuit designs. These metrics quantitatively measure the model's generation performance in preserving critical design rules and geometric precision.

Specifically, we define four fundamental design rules: Circuit Linkage Constraint (CLC), Power Delivering Constraint (PDC), Horizontal Spacing Constraint (HSC), and Vertical Spacing Constraint (VSC). CLC measures the overall functionality of the layout, while PDC, HSC, VSC measure whether device layers are placed correctly. To quantify performance, we measured the non-functional (overlapped-circuit) area for CLC and violation rates by counting layers that break for the other three rules. Detailed explanations of metrics are provided through equations:

---

[1]Upon acceptance, we will publicly release the dataset and evaluation code under CC BY-NC-SA 4.0.

1. **Power Delivery Constraint:** Layer 1457 must reside above Layer 515 to ensure adequate power delivery.

$$\forall_i : \quad \text{Area}\big(L_{515} \cap L_{1457}^{(i)}\big) > 0 \tag{16}$$

2. **Circuit Linkage Constraint:** Layer 644 and Layer 515 must not overlap each other to maintain overall functionality of the circuit.

$$\forall_{i \neq j} : \text{Area}\big(\big(L_{515} \cap L_{644}\big) \cup \big(L_{515}^{(i)} \cap L_{515}^{(j)}\big) \cup \big(L_{644}^{(i)} \cap L_{644}^{(j)}\big)\big) = 0 \tag{17}$$

3. **Horizontal Spacing Constraint:** Instances of Layer 1457 aligned vertically must maintain a minimum horizontal separation of $W$.

$$\forall_{i \neq j} \; if \; |L_{1457,y}^{(i)} - L_{1457,y}^{(j)}| < \epsilon : \quad |L_{1457,x}^{(i)} - L_{1457,x}^{(j)}| \geq W \tag{18}$$

4. **Vertical Spacing Constraint:** Instances of Layer 1457 aligned horizontally must maintain a minimum vertical separation of $H$.

$$\forall_{i \neq j} \; if \; |L_{1457,x}^{(i)} - L_{1457,x}^{(j)}| < \epsilon : \quad |L_{1457,y}^{(i)} - L_{1457,y}^{(j)}| \geq H \tag{19}$$

Here, Area and Layer (L) represent the computational areas and layer components, respectively, from the ContLayNet dataset at a real-world scale. We normalize each metric to $[0, 1]$ by dividing the number of violating components by the maximum possible violations. For Horizontal and Vertical Spacing Constraints, which measure the minimum pairwise distance between Device (Layer 1457) layers, we add a penalty for samples with fewer than a given threshold number of Device layers. This prevents models from artificially improving spacing scores by simply omitting critical components, as layouts with fewer Device layers can trivially satisfy large minimum-distance requirements. For HSC and VSC, $\epsilon, W, H$ are set as 240, 1200, and 1000 during the experiments.

## 6 Experiments

This section presents the experimental setup and evaluates the performance of AGDC compared to existing methods across three domains: semiconductor layouts, graphic layouts, and SVGs.

### 6.1 ContLayNet

**Baselines.** We compare AGDC with two representative baselines: LayoutTransformer (LT) (Gupta et al., 2021), an autoregressive model tailored for sequential data generation but limited by discretization, and DLT (Levi et al., 2023), a non-autoregressive diffusion model specifically designed for graphic layout generation that jointly models discrete and continuous values. Matching our hardware, we evaluate LT with 18-bit precision (the maximum precision achievable with a batch size of 2 on a single NVIDIA A6000 GPU), while retaining hybrid vector representations for both DLT and AGDC.

**Implementation details.** AGDC employs a Transformer decoder architecture with a hidden dimension of 1024, comprising 32 decoder layers and 16 attention heads. $\text{MLP}_{\text{CONT}}$ consists of three blocks with 1024 channels each, while $\text{MLP}_{\text{DISC}}$ and $\text{MLP}_{\text{EOS}}$ are both two-layer MLPs with hidden layers twice the width of their input dimensions. We set $\lambda_1 = 100$, $\lambda_2 = 0.1$, and $\alpha = 0.1$. Additional implementation details and inference cost analysis are provided in the appendix.

**Results.** Table 2 presents the performance of AGDC on the ContLayNet benchmark. Across both completion tasks with 50 and 100 layers, AGDC outperforms all baselines on the four metrics CLC, PDC, HSC, and VSC. These benchmark results demonstrate the clear superiority of our proposed framework.

Figure 5 illustrates the qualitative performance of AGDC. LT exhibits oversized layers that overlap large portions of the layout, often obscuring underlying structures. DLT tends to generate many noisy elements due to poor length control capabilities (Figure B). In contrast, AGDC produces more balanced and well-distributed layers than LT and DLT, showing encouraging alignment with the real samples while offering opportunities for further improvement.

Table 2: **Comparison on the ContLayNet Benchmark.** AGDC outperforms baseline methods across CLC, HSC, and VSC metrics. Ablation studies demonstrate the contribution of both EOS logit adjustment mechanism and length regularization components. Bold indicates best performance.

| Precision | Completion (given 50 layers) | | | | Completion (given 100 layers) | | | |
|---|---|---|---|---|---|---|---|---|
| | CLC ↓ | PDC ↓ | HSC ↓ | VSC ↓ | CLC ↓ | PDC ↓ | HSC ↓ | VSC ↓ |
| 18 (LT) | 0.397 | 0.149 | 0.053 | 0.058 | 0.152 | 0.144 | 0.051 | 0.054 |
| ∞ (DLT) | 0.227 | 0.125 | 0.065 | 0.056 | 0.144 | 0.119 | 0.065 | 0.052 |
| ∞ (AGDC) | **0.088** | **0.068** | **0.027** | **0.023** | **0.046** | **0.058** | **0.024** | **0.025** |
| - w/o $\text{MLP}_{\text{EOS}}$ | 0.101 | 0.078 | 0.028 | 0.034 | 0.050 | 0.092 | 0.027 | 0.033 |
| - w/o $\mathcal{L}_\ell$ | 0.107 | 0.076 | 0.032 | 0.038 | 0.067 | 0.068 | 0.031 | 0.032 |
| - w/o both | 0.101 | 0.113 | 0.030 | 0.029 | 0.056 | 0.139 | 0.031 | 0.034 |
| Real | 0.0 | 0.0 | 0.0 | 0.0 | 0.0 | 0.0 | 0.0 | 0.0 |

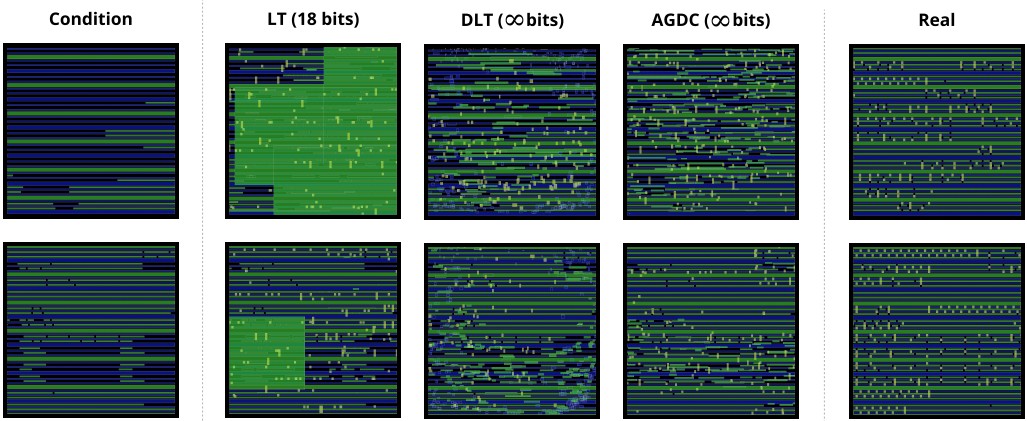

Figure 5: **Qualitative results on ContLayNet in completion task given 50 / 100 layers.** AGDC illustrate clearly superior performance to LT and DLT.

**Ablation studies.** Table 2 also includes ablation studies for our two key components: the EOS logit adjustment mechanism ($\text{MLP}_{\text{EOS}}$) and length regularization term ($\mathcal{L}_\ell$). Results demonstrate that both components contribute to performance, with the combination achieving the best results across every metric. Figure 6 further illustrates the impact of these components on length control by showing the fitted Gaussian distributions of length errors (sampled − ground truth). Our complete AGDC framework achieves a near-zero mean error ($\mu = 0.15$) with reduced variance ($\sigma = 146.13$), while the ablated version exhibits both bias ($\mu = 38.91$) and higher variance ($\sigma = 154.45$), confirming the effectiveness of our components.

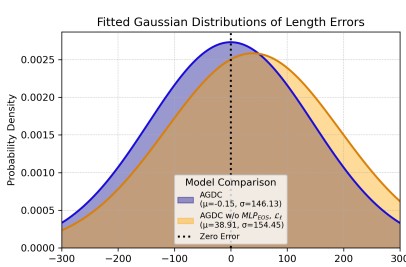

Figure 6: **Length error comparisons.**

## 6.2 GRAPHIC LAYOUT GENERATION

**Experimental setup.** We use *PubLayNet* (Zhong et al., 2019) and *Rico* (Deka et al., 2017), which contain 330K scientific document layouts and 91K mobile UI layouts, respectively. For evaluation, we employ three standard metrics: *Fréchet inception distance* (FID) (Heusel et al., 2017), *Overlap* (Li et al., 2019), and *Alignment* score (Lee et al., 2020). We compare against the same baseline models as in Section 6.1: LayoutTransformer (Gupta et al., 2021) with precision 18 bits, and DLT (Levi et al., 2023). Additional details on datasets, metrics, baselines, and implementation specifics are available in the appendix.

**Results.** On both PubLayNet and Rico datasets, AGDC outperforms both LT (18 bits) and DLT across all tasks, including completion and unconditioned generation (Table 3). AGDC achieves the

Table 3: **Comparison on high-precision graphic layout generation.** AGDC consistently outperforms baselines across all metrics, with particularly significant gains on PubLayNet. Ablation studies confirm the effectiveness of our key components across datasets. Metrics: *FID* (lower is better), *Overlap* and *Alignment* scores (closer to real is better, $\times 100$ for clarity).

| | **PubLayNet** | | | | | | **Rico** | | | | | |
| | Completion | | | Un-Gen | | | Completion | | | Un-Gen | | |
| Precision | FID | Overlap | Align | FID | Overlap | Align | FID | Overlap | Align | FID | Overlap | Align |
| 18 (LT) | 186.61 | 64.16 | 1.06 | 198.76 | 59.97 | 1.91 | 24.32 | 45.95 | 0.35 | 28.29 | 50.84 | 0.41 |
| $\infty$ (DLT) | 35.80 | 9.65 | 0.84 | 44.00 | 8.64 | 0.71 | 20.92 | 28.11 | 0.63 | 38.97 | 38.87 | 0.39 |
| $\infty$ (**AGDC**) | **4.58** | **4.59** | **0.17** | **12.36** | **4.00** | **0.20** | 9.77 | **33.11** | **0.25** | **28.19** | **31.90** | **0.30** |
| - w/o MLP$_{EOS}$ | 5.27 | 5.08 | **0.17** | 12.75 | 4.79 | 0.21 | 10.16 | 32.05 | 0.26 | 30.40 | 31.63 | 0.32 |
| - w/o $\mathcal{L}_\ell$ | 5.05 | 4.65 | 0.18 | 12.51 | 4.67 | 0.21 | **9.63** | 32.07 | 0.25 | 28.32 | 30.75 | 0.35 |
| - w/o both | 5.07 | 5.25 | 0.18 | 12.90 | 5.00 | 0.21 | 10.76 | 31.62 | 0.29 | 29.29 | 29.74 | 0.36 |
| Real | – | 0.22 | 0.03 | – | 0.22 | 0.03 | – | 32.92 | 0.17 | – | 32.92 | 0.17 |

Figure 7: **Top: Layout Generation on PubLayNet. Bottom: Text-to-SVG Generation on FIGR-8-SVG.** AGDC achieves superior performance in high-precision settings while maintaining quality comparable to low-precision settings across diverse domains.

best FID, Overlap, and Alignment scores, demonstrating its superiority in high-precision settings, where LT and DLT face significant challenges. Ablation studies on these datasets confirm the effectiveness of both the EOS logit adjustment mechanism and length regularization components, with consistent improvements observed across metrics. Figure 7 demonstrates the qualitative performance of AGDC, which achieves comparable quality to low-precision LT while outperforming high-precision LT and DLT. Additional qualitative results, including unconditioned generation and experiments on the Rico dataset, are provided in the appendix.

## 6.3 TEXT-TO-SVG GENERATION

**Experimental setup.** To our knowledge, we evaluate AGDC as the first method to preserve the continuous nature of text-to-SVG generation, comparing it against IconShop (Wu et al., 2023)—the only other text-to-SVG method operating at the control-point level—in both high and low-precision settings. For our experiments, we use *FIGR-8-SVG* dataset (Clouâtre & Demers, 2019), which contains 1.5M SVG-formatted black-and-white icons, following IconShop's pre-processing protocol.

We employ two standard metrics: *Fréchet inception distance* (FID) (Heusel et al., 2017) and *CLIP score* (Radford et al., 2021). Additional details on datasets, metrics, and implementation specifics are available in the appendix.

**Results.** Table 4 and Figure 7 demonstrate AGDC's superior performance in high-precision settings. While Icon-Shop produces visible results at 8 bits but degrades at 9 bits and fails completely beyond 10 bits, AGDC generates continuous vector outputs with inherent infinite precision that maintain quality comparable to IconShop's best outputs. Ablation studies further confirm that both key components ($\text{MLP}_{\text{EOS}}$ and $\mathcal{L}_\ell$) contribute meaningfully to the enhanced performance.

Table 4: **Text-to-SVG results.**

| Precision | Model | FID $\downarrow$ | CLIP $\uparrow$ |
|---|---|---|---|
| 8 | IconShop | 37.35 | 22.13 |
| 9 | IconShop | 95.96 | 18.07 |
| 10 | IconShop | Fail | Fail |
| $\infty$ | AGDC | 48.73 | 20.05 |
| $\infty$ | - w/o both | 58.22 | 19.90 |

## 7 CONCLUSION

This paper introduces AGDC, a novel autoregressive framework that jointly models discrete and continuous values within variable-length sequences, addressing fundamental limitations of discretization-based approaches. Our key technical components include an EOS control mechanism with MLP-based logit adjustment and a length regularization term, both of which contribute to optimal performance. We present ContLayNet, a large-scale benchmark comprising 334K high-precision semiconductor layouts with specialized DRC metrics, addressing the critical gap in evaluation datasets where precision errors directly impact functional performance. Experiments across semiconductor layouts, graphic layouts, and text-to-SVG synthesis demonstrate AGDC's superior performance in high-precision scenarios, outperforming existing discretization-based and fixed-schema methods.

Despite our promising results, challenges remain for future exploration. While AGDC outperforms existing methods, all approaches, including ours, struggle with the complexity of ContLayNet, particularly for longer sequences, where error propagation becomes more pronounced. This highlights both the difficulty of the task and opportunities for future work. Our work advances hybrid vector representation learning for domains requiring both semantic richness and geometric precision, with ContLayNet serving as a valuable resource for evaluating future approaches in high-precision engineering applications. See Section A.4 for the applications and impact of generated circuits.

## ETHICS STATEMENT

This work presents AGDC, a general framework for generating hybrid discrete-continuous sequences, and ContLayNet, a semiconductor layout dataset containing no personal or proprietary information that will be made publicly available. While our experiments focus on legitimate domains (semiconductor layouts, graphic layouts, SVG), the general nature of our framework could potentially be misused for creating deceptive content or automating malicious design processes. We acknowledge these dual-use risks and encourage researchers to exercise appropriate caution, while believing the benefits of advancing high-precision generative modeling for engineering applications outweigh the potential risks given responsible usage.

## REPRODUCIBILITY STATEMENT

To ensure reproducibility, we provide complete experiment details for all three domains in the Appendix: ContLayNet: Section A.1, Graphic Layout Generation: Section B.2, Text-to-SVG: Section C.1), including hyperparameters and implementation specifics. We include anonymous source code as supplementary materials containing our framework implementation and experimental scripts to reproduce all reported results across all domains.

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

# A CONTLAYNET: SEMICONDUCTOR LAYOUT GENERATION

## A.1 EXPERIMENT DETAILS

**Implementation details.** AGDC utilizes a Transformer decoder architecture with 32 layers, 16 attention heads, and a hidden dimension of 1024. For the denoising process, we employ an MLP comprising three blocks, each with 1024 channels, following the diffusion approach of (Nichol & Dhariwal, 2021). During training, each latent vector $\mathbf{z}$ is processed through the denoising MLP, sampling the timestep $t$ 30 times each. $\text{MLP}_{\text{CONT}}$ consists of three blocks with 1024 channels each, while $\text{MLP}_{\text{DISC}}$ and $\text{MLP}_{\text{EOS}}$ are both two-layer MLPs with hidden layers twice the width of their input dimensions. We set $\lambda_1 = 100$, $\lambda_2 = 0.1$ and $\alpha = 0.1$. The model was trained for 10 days on a single NVIDIA A6000 GPU with a learning rate of $7.5 \times 10^{-5}$. Since the ContLayNet has been newly introduced, further optimization and refinement may yield improved performance.

## A.2 INFERENCE COSTS AND ACCELERATION

Table A presents comprehensive timing comparisons for the completion task with 50 layers, sampling 100 samples with a batch size of 5 on a single RTX 4090 GPU. Throughout this paper, AGDC uses Improved DDPM (Nichol & Dhariwal, 2021) with 100 diffusion steps for all reported results unless otherwise specified. To address inference speed concerns, we evaluate sampling acceleration by reducing diffusion steps. Improved DDPM's learned variance formulation enables effective sampling with fewer steps using uniform stride sequences. We also evaluate DDIM (Song et al., 2021) as an alternative sampler, which uses a deterministic non-Markovian process that also allows timestep skipping. We also compare against baseline methods LT and DLT.

Table A: **Inference time and generation quality comparison across methods.** AGDC with Improved DDPM at 50 steps achieves comparable speed to LT while substantially outperforming on all DRC metrics. AGDC demonstrates favorable speed-quality trade-offs compared to LT and DLT across different sampling configurations. Bold indicates best performance.

| Model | Sampling Method | steps | Time (s) | CLC ↓ | PDC ↓ | HSC ↓ | VSC ↓ |
|-------|-----------------|-------|----------|-------|-------|-------|-------|
| LT | - | - | 766.12 | 0.375 | 0.148 | 0.049 | 0.049 |
| DLT | - | - | 7.64 | 0.202 | 0.179 | 0.070 | 0.053 |
| AGDC | Improved DDPM | 100 | 1489.13 | **0.084** | **0.067** | **0.026** | **0.022** |
| | | 50 | 876.78 | 0.098 | 0.082 | 0.034 | 0.023 |
| | | 20 | 616.88 | 0.114 | 0.090 | 0.035 | 0.029 |
| | | 10 | 579.97 | 0.179 | 0.088 | 0.097 | 0.057 |
| | DDIM | 100 | 1800.44 | 0.090 | 0.085 | 0.032 | 0.027 |
| | | 50 | 1205.35 | 0.107 | 0.085 | 0.032 | 0.031 |
| | | 20 | 888.95 | 0.139 | 0.117 | 0.056 | 0.045 |
| | | 10 | 721.95 | 0.210 | 0.127 | 0.123 | 0.080 |

Table A demonstrates that AGDC achieves practical inference speeds while delivering superior generation quality. With Improved DDPM sampling at 50 steps, AGDC runs at a comparable speed to LT while substantially outperforming across all DRC metrics. Further acceleration is possible by reducing diffusion steps, though with some performance degradation: at 20 steps, AGDC becomes faster than LT while maintaining quality advantages. We also evaluated DDIM as an alternative sampler, but it showed both slower inference times and lower quality compared to Improved DDPM at equivalent step counts. While DLT achieves the fastest inference through its non-autoregressive architecture, its non-autoregressive nature makes it difficult to control sequence length, resulting in many noisy elements, particularly on variable-length datasets like ContLayNet (Figure B). These results demonstrate that users can reduce inference time at the cost of generation quality based on their preference, while still maintaining substantially better functional correctness than discretization baselines even at reduced step counts.

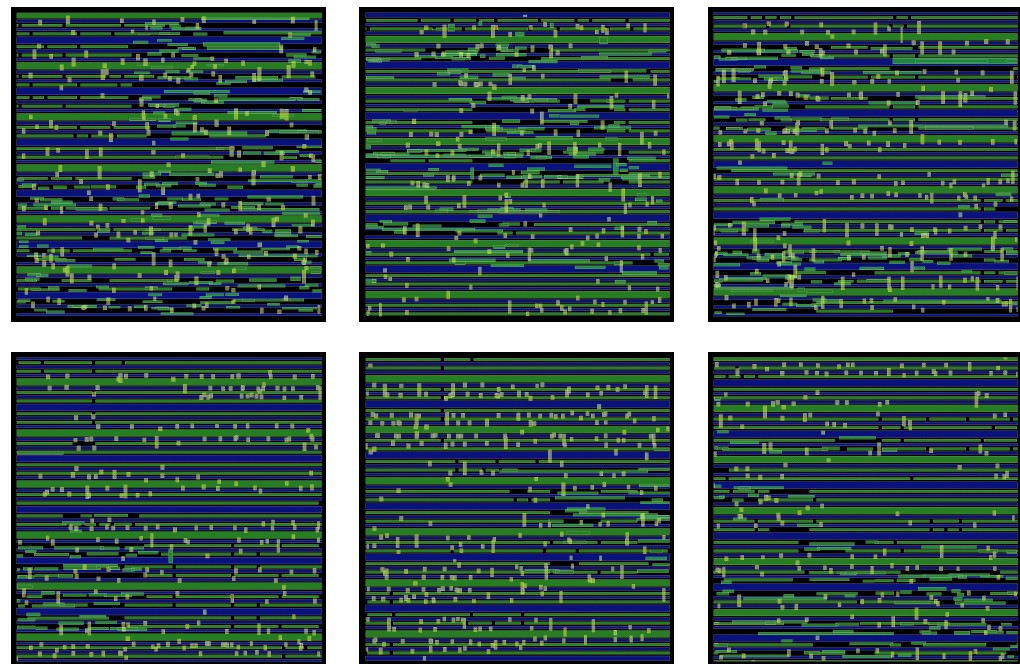

Figure A: **Qualitative results on ContLayNet.** AGDC produces more balanced and well-distributed layers than baseline methods, offering opportunities for future improvement.

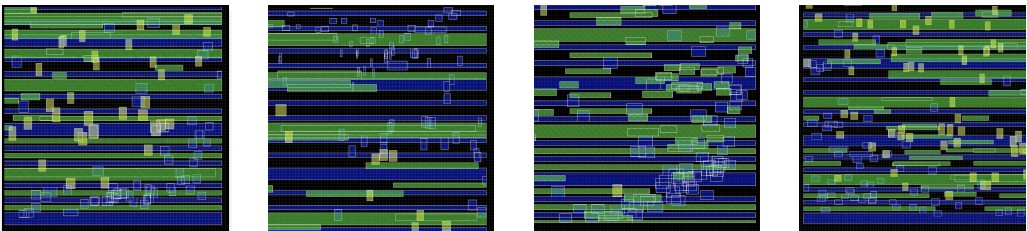

Figure B: **Magnified DLT results on ContLayNet.** DLT generates numerous small, noisy elements due to poor length control in its non-autoregressive architecture.

## A.3   ADDITIONAL RESULTS.

Figure A shows additional samples from AGDC, producing more balanced and well-distributed layers than baseline methods. Figure B shows magnified views of DLT outputs, revealing numerous small, noisy elements scattered throughout the layouts. This over-generation stems from DLT's lack of capability to control sequence termination in its non-autoregressive setting, which is particularly problematic for variable-length datasets like ContLayNet.

## A.4   PRACTICAL APPLICATIONS AND INDUSTRIAL IMPACT.

The generated circuits from our approach offer several practical applications in semiconductor layout design. Generated layouts provide manufacturable starting configurations for design engineers, substantially reducing design space exploration overhead while ensuring adherence to manufacturing constraints. Our approach systematically identifies novel layout configurations that satisfy manufacturing requirements while exploring previously inaccessible regions of the feasible design space. The model generates both compliant and non-compliant samples, enabling comprehensive analysis of constraint violations and potential failure modes critical for robust layout synthesis.

# B  GRAPHIC LAYOUT GENERATION

## B.1  EMPIRICAL ANALYSIS OF PRECISION LIMITATIONS

Table B demonstrates how precision affects LayoutTransformer (LT) performance across both Pub-LayNet and Rico datasets. The results confirm our theoretical analysis in Section 3.1: as precision increases, the exponential growth in token count leads to significant performance degradation. At low precision (4 bits), models exhibit poor performance with high FID scores due to insufficient representational capacity. Performance peaks at moderate precision levels (8-12 bits), where the model achieves an optimal balance between representational power and computational feasibility. However, at higher precision (16 bits), performance deteriorates markedly due to training instability caused by vocabulary explosion. This pattern consistently appears across both datasets and all generation tasks, highlighting a fundamental limitation of discretization-based methods in high-precision settings.

Table B: **Empirical Analysis of Precision Limitations.** Performance significantly degrades at higher levels (16 bits) due to vocabulary explosion. Metrics: *FID* (lower is better), *Overlap* and *Alignment* scores (closer to real is better, $\times 100$ for clarity).

| Dataset | PubLayNet | | | | | | Rico | | | | | |
|---|---|---|---|---|---|---|---|---|---|---|---|---|
| Task | Completion | | | Un-Gen | | | Completion | | | Un-Gen | | |
| Precision | FID | Overlap | Align. | FID | Overlap | Align. | FID | Overlap | Align. | FID | Overlap | Align. |
| 4 | 49.25 | 5.25 | 0.15 | 55.49 | 4.91 | 0.17 | 87.13 | 33.78 | 0.32 | 86.96 | 34.50 | 0.31 |
| 8 | 3.00 | 2.78 | 0.08 | 11.12 | 2.50 | 0.13 | 5.20 | 53.47 | 0.21 | 6.00 | 52.40 | 0.20 |
| 12 | 12.93 | 7.72 | 0.13 | 22.51 | 8.69 | 0.19 | 8.90 | 58.22 | 0.29 | 10.53 | 58.79 | 0.26 |
| 16 | 165.63 | 57.08 | 0.75 | 178.68 | 52.69 | 1.42 | 20.35 | 50.17 | 0.39 | 22.08 | 51.44 | 0.30 |
| Real | - | 0.22 | 0.03 | - | 0.22 | 0.03 | - | 58.96 | 0.17 | - | 58.96 | 0.17 |

## B.2  EXPERIMENT DETAILS

**Datasets.**  *PubLayNet* (Zhong et al., 2019) contains 330K scientific document layouts with five component types: text, title, figure, list, and table. *Rico* (Deka et al., 2017) comprises 91K mobile UI layouts with 27 component types. We focused on the top 13 component types, following the approach of (Lee et al., 2020). For both datasets, we limited our analysis to layouts containing 9 or fewer components, consistent with methods (Kikuchi et al., 2021; Levi et al., 2023). We used the train-test split defined by (Kikuchi et al., 2021).

**Metrics.**  We evaluate layout quality using three common metrics: *Fréchet inception distance* (FID) (Heusel et al., 2017), which measures the similarity between generated and real layouts; *Overlap* (Li et al., 2019), which quantifies the total overlapping area between components; and *Alignment* score (Lee et al., 2020), which evaluates how well components are aligned. We implement all of these metrics following LayoutGAN++ (Kikuchi et al., 2021), utilizing the same pre-trained feature extraction model.

**Implementation details.**  We adopt a Transformer decoder architecture with a hidden dimension of 1024, 6 decoder layers, and 8 attention heads. The denoising MLP consists of 6 blocks with 1024-channel width, and the diffusion process follows (Nichol & Dhariwal, 2021). During training, we sample the timestep $t$ 32 times for each latent vector $\mathbf{z}$ of the denoising MLP. For PubLayNet's discrete branch and for EOS logit adjustment, we employ a two-layer MLP with GELU activation and dropout, using a hidden layer twice the input width. Due to the training instability with smaller datasets, we use a simpler linear layer for Rico instead of the MLP. The models were trained on PubLayNet (learning rate $7.5 \times 10^{-5}$ and Rico (learning rate $1.5 \times 10^{-4}$), both completing within 15 hours on a single RTX 4090. We set $\lambda_1 = 100$, $\lambda_2 = 0.005$, and $\alpha = 0.05$.

## B.3  ADDITIONAL RESULTS

**PubLayNet.**  Figure C and Figure D present additional qualitative results for the completion task and unconditioned generation task on PubLayNet, respectively. AGDC achieves comparable structural

coherence to low-precision LT while maintaining higher precision capabilities. Each rectangular box represents a region of a specific document element (*e.g.,* text, title, figure), with different colors indicating different element classes.

**Rico.** Figure E shows results for the completion task on Rico, where AGDC effectively preserves layout consistency and element relationships. Figure F demonstrates unconditioned generation results, with generated layouts maintaining appropriate mobile UI design patterns.

## C TEXT-TO-SVG GENERATION

### C.1 EXPERIMENT DETAILS

**Text-to-SVG implementation.** Following IconShop (Wu et al., 2023), text descriptions are tokenized using a pretrained BERT encoder and prepended to the SVG sequence for joint autoregressive modeling. The model learns to map textual semantics to appropriate SVG commands and coordinates through end-to-end training, generating vector graphics that align with input text descriptions.

**Datasets.** *FIGR-8-SVG* (Clouâtre & Demers, 2019) is composed of 1.5M samples of SVG-formatted black-and-white icons. The preprocessing of the data followed the procedure of Iconshop (Wu et al., 2023) to get the valid text-SVG pairs. We excluded paired data with text longer than 50 words or SVG longer than 256 atomic units. For the training validation and testing of the models, 72K samples each are extracted from the data set as validation set and test set.

**Metrics.** We evaluate SVG generation quality using two commonly adopted metrics: *Fréchet inception distance* (FID) (Heusel et al., 2017), which measures the similarity between the generated SVG and real SVG in rendered image space and *CLIP score* (Radford et al., 2021), which evaluates the similarity score between the generated rendered SVG icon and input text condition.

**SVG representation.** Following IconShop (Wu et al., 2023), each SVG atomic unit is represented with a standardized 8-dimensional continuous vector (four coordinate pairs: start point, control point 1, control point 2, and end point). Different command types use only relevant coordinates and ignore unused ones. This unified representation enables consistent tensor operations across all command types while handling their varying coordinate requirements.

**Implementation details.** We adopt a Transformer decoder architecture with 8 decoder layers, 8 attention heads, and a hidden dimension of 1024. For the denoising process, we employ an MLP comprising three blocks, each with 256 channels, following the diffusion approach of (Nichol & Dhariwal, 2021). During training, each latent vector $\mathbf{z}$ is processed through the denoising MLP, sampling the timestep $t$ 4 times each. The discrete branch for SVG generation consists of a two-layer MLP with a hidden layer same as the input width, using RELU activation and dropout. we set $\lambda_1 = 100$, $\lambda_2 = 0.001$, and $\alpha = 0.005$. The model was trained on FIGR-8-SVG for 10 days on $4 \times$ NVIDIA A6000 GPUs, with a learning rate of $1.0 \times 10^{-4}$.

## D USE OF LARGE LANGUAGE MODELS

AI-powered language models provided editorial support for linguistic refinement, including grammatical corrections, syntactic enhancement, and stylistic improvements across the document. The substantive content, research findings, experimental approaches, and scholarly assertions represent solely the authors' original contributions.

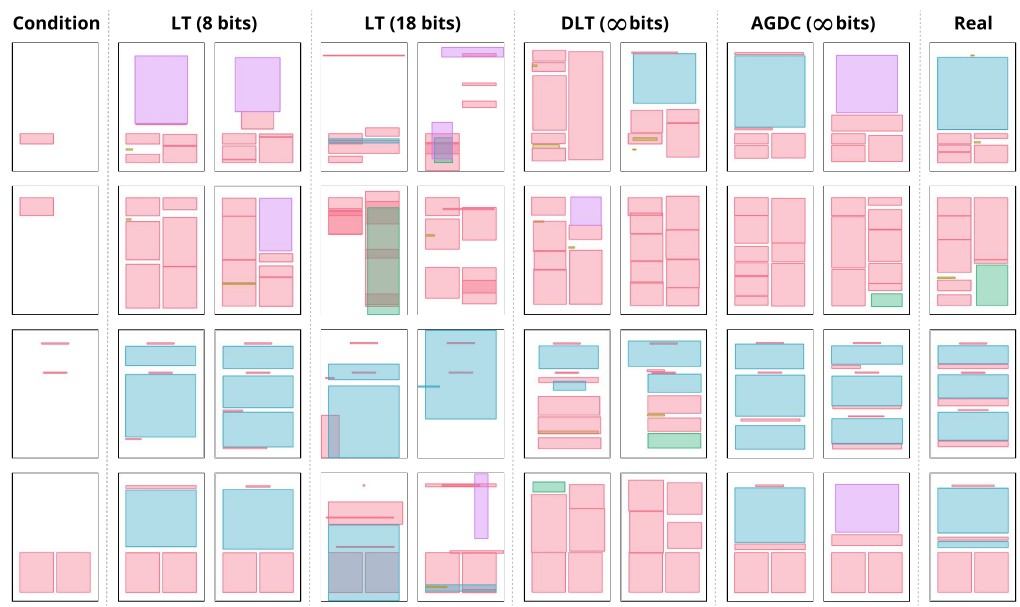

Figure C: **Qualitative results on PubLayNet, Completion task.** AGDC (Ours) achieves comparable quality to low-precision LT while outperforming high-precision LT and DLT.

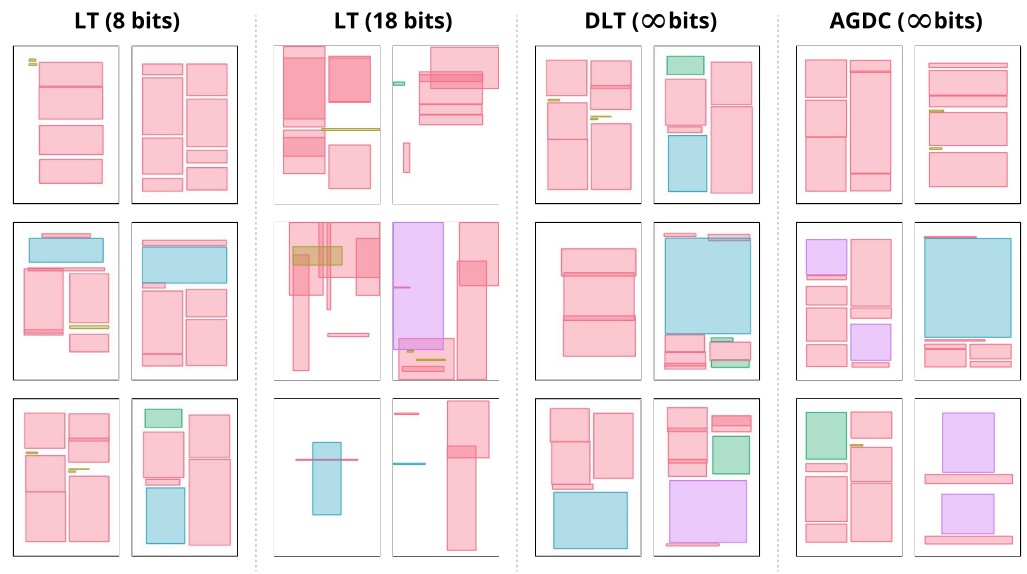

Figure D: **Qualitative results on PubLayNet, Un-Gen task.** AGDC (Ours) achieves comparable quality to low-precision LT while outperforming high-precision LT and DLT.

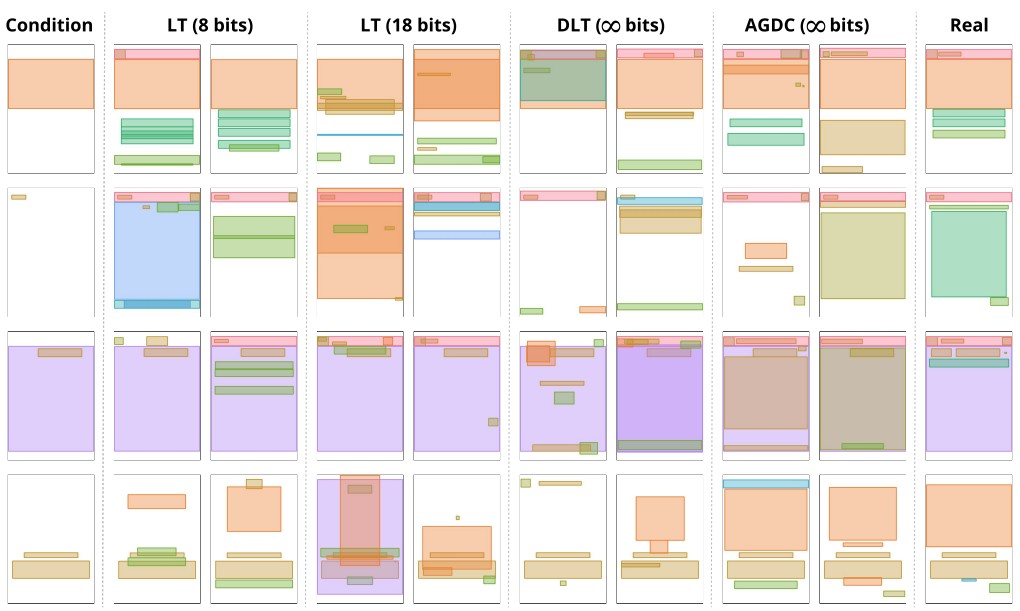

Figure E: **Qualitative results on Rico, Completion task.** AGDC (Ours) achieves comparable quality to low-precision LT while outperforming high-precision LT and DLT.

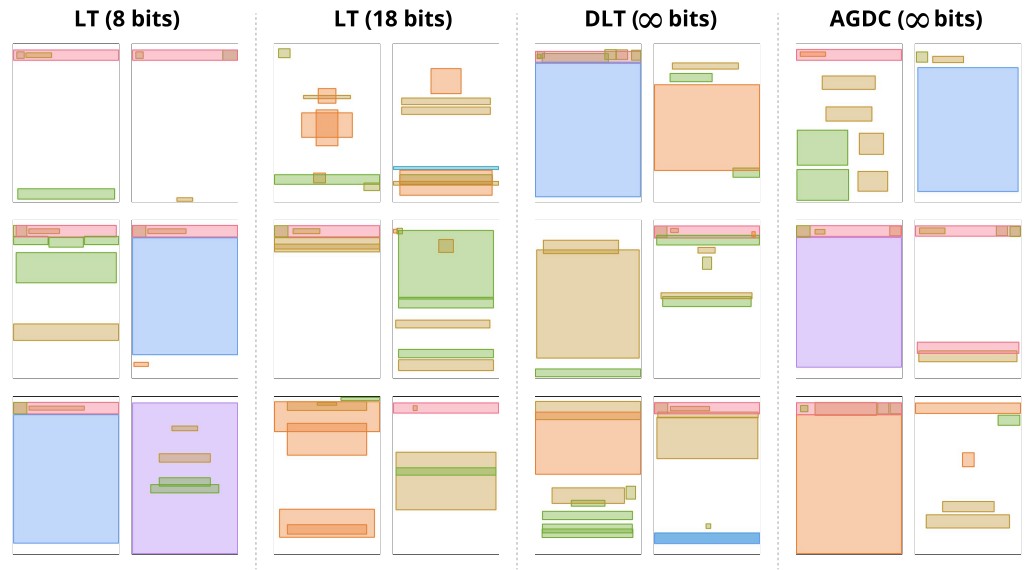

Figure F: **Qualitative results on Rico, Un-Gen task.** AGDC (Ours) achieves comparable quality to low-precision LT while outperforming high-precision LT and DLT.

