# OpenReview forum: "AGDC: Autoregressive Generation of Variable-Length Sequences with Joint Discrete and Continuous Spaces"
_ICLR.cc/2026/Conference — Submitted to ICLR 2026_

### Official Review · Reviewer_r7X6 · 2025-10-28

**Soundness:** 2
**Presentation:** 2
**Contribution:** 2
**Rating:** 4
**Confidence:** 2

**Summary:**

The paper proposes AGDC, a novel autoregressive framework for generating variable-length sequences that contain both discrete identifiers and high-precision continuous values. The core problem it addresses is the precision loss inherent in standard tokenization-based models, which is detrimental in domains like semiconductor design. AGDC tackles this by jointly modeling the two data types: using categorical prediction for discrete values and a conditional diffusion model for continuous vectors, all within a unified autoregressive structure. The authors also introduce ContLayNet, a large-scale benchmark of high-precision semiconductor layouts, and a corresponding set of Design Rule Check (DRC) metrics to evaluate functional correctness.

**Strengths:**

1. The paper identifies a critical and practical limitation of discretization in generative models. The proposed hybrid approach of combining autoregressive prediction with an inner loop of diffusion sampling is an elegant and novel method to preserve continuous precision.
2. The introduction of the ContLayNet benchmark and its specialized DRC-based evaluation metrics is a major contribution. This provides a much-needed and challenging testbed for a problem space that has been underserved.

**Weaknesses:**

1. A major concern is the inference speed. The model must run an iterative diffusion sampling process (which is itself multi-step) for every single autoregressive step. This seems computationally prohibitive and likely orders of magnitude slower than discretization-based autoregressive models, potentially limiting its practical utility.
2. It remains unclear what this paper contributes to the machine learning (generative models) community.

**Questions:**

Could the authors provide a more direct comparison of inference time between AGDC, LT, and DLT for generating, for example, 100 samples? How many sampling steps does the diffusion model use per autoregressive step, and was acceleration (e.g., DDIM) explored?

---

> ### Author Response · Authors · 2025-11-18
> **Response to Reviewer r7X6 (Part 1)**
>
> We appreciate the reviewer’s acknowledgment that our hybrid approach is “elegant and novel” for preserving continuous precision, and that ContLayNet provides “a much-needed and challenging testbed.” We sincerely thank the reviewers for their insightful comments and constructive suggestions, which have significantly improved the quality of our paper. We address each concern below.
>
> **[W1 & Q1] Concerns on Inference Speed**
>
> We provide more comprehensive experimental results, including various options for the inference speed of AGDC in Table R1. Note that we managed to optimize the speed of LayoutTransformer (LT), achieving ~3$\times$ speed-up without losing accuracy. As shown in Table R1, AGDC with 20 steps is still faster than LT by ~15%, and, at the same time, outperforms LT significantly in all metrics. While DLT achieves fast inference through its non-autoregressive architecture, its non-autoregressive nature makes it challenging to control sequence length, resulting in many noisy elements, particularly on variable-length datasets like ContLayNet. Note that DLT is worst in most cases according to our experiments. (See magnified DLT results in Figure B in the updated Appendix A.3 and check FID scores in our response to Reviewer 65Z6.)
>
> When we compare our approach (AGDC) based on improved DDPM [R15] with DDIM [R16], AGDC is noticeably and consistently better regardless of the number of steps in most metrics, except in one case (HSC with 50 steps).
>
> **Table R1**: Inference time and generation quality comparison across methods.
> | Model | Method | # of steps | Time (s) | CLC $\downarrow$ |  PDC $\downarrow$ | HSC $\downarrow$ | VSC $\downarrow$ |
> |-------|-----------------|-----------|----------|------------------|-------------------|------------------|------------------|
> | LT    | -               | -         | 766.12   | 0.375            | 0.148             | 0.049            | 0.049            |
> | DLT   | -               | -         | 7.64     | 0.202            | 0.179             | 0.070            | 0.053            |
> | AGDC  | Improved DDPM   | 100       | 1489.13  | 0.084            | 0.067             | 0.026            | 0.022            |
> |       |                 | 50        | 876.78   | 0.098            | 0.082             | 0.034            | 0.023            |
> |       |                 | 20        | 616.88   | 0.114            | 0.090             | 0.035            | 0.029            |
> |       |                 | 10        | 579.97   | 0.179            | 0.088             | 0.097            | 0.057            |
> |       | DDIM            | 100       | 1800.44  | 0.090            | 0.085             | 0.032            | 0.027            |
> |       |                 | 50        | 1205.35  | 0.107            | 0.085             | 0.032            | 0.031            |
> |       |                 | 20        | 888.95   | 0.139            | 0.117             | 0.056            | 0.045            |
> |       |                 | 10        | 721.95   | 0.210            | 0.127             | 0.123            | 0.080            |
>
> [R15] Alex Nichol and Prafulla Dhariwal. Improved denoising diffusion probabilistic models. In ICML, 2021.
>
> [R16] Jiaming Song, Chenlin Meng, and Stefano Ermon. Denoising diffusion implicit models. In ICLR, 2021.

---

> ### Author Response · Authors · 2025-11-18
> **Response to Reviewer r7X6 (Part 2)**
>
> **[W2] Contribution to the machine learning community**
>
> We would like to clarify how our work contributes to the broader machine learning community.
>
> **Methodological Contribution.**
> AGDC introduces a unified autoregressive framework for jointly modeling discrete and continuous values in variable-length sequences, which is a fundamental challenge across diverse domains. Although autoregressive models for discrete tokens and diffusion models for continuous data have been studied independently, their integration into a single architecture that supports hybrid generation with explicit control over sequence length is, to our knowledge, novel. Our key technical contributions include: (1) unified latent conditioning for both categorical and diffusion-based generation, (2) an MLP-based EOS logit adjustment module for precise length control, and (3) a differentiable length regularization objective enabling end-to-end optimization of sequence length.
>
> **Benchmark Contribution.**
> ContLayNet fills a critical gap by providing a dataset where precision errors translate directly into functional failures. Unlike conventional perceptual metrics (*e.g.*, FID), our DRC metrics explicitly measure how precision loss affects correctness. This allows systematic comparison between discretization-based and continuous-generation methods and offers a valuable evaluation framework for researchers developing high-precision generative models.
>
> **Broader Impact.**
> As the reviewer notes, this problem space has been underserved. Generating sequences that contain both discrete and continuous components arises in many domains—such as semiconductor layouts, graphic layouts, and SVG generation in our experiments, as well as molecular design [R17], music generation [R18], and CAD modeling [R19] beyond the scope of this paper. By introducing a domain-agnostic framework together with a rigorous evaluation benchmark, our work provides a general solution for these underexplored application areas and contributes broadly to the machine learning community.
>
> [R17] Alex Morehead and Jianlin Cheng. Geometry-complete diffusion for 3D molecule generation and optimization. In ICLR Workshop, 2023.
>
> [R18] Cheng-Zhi Anna Huang, et al. Music Transformer: Generating music with long-term structure, In ICLR, 2019.
>
> [R19] Xiang Xu, et al. SkexGen: Autoregressive Generation of CAD Construction Sequences with Disentangled Codebooks, In ICML, 2022.

---

> ### Author Response · Authors · 2025-11-24
> **Request for Reviewer Engagement - Paper 5060**
>
> Dear Reviewer r7X6,
>
> Thank you for coordinating the review process for ICLR 2026.
>
> We have submitted a comprehensive rebuttal addressing all of your concerns, but unfortunately have not yet received responses. We believe that engaging in dialogue with the reviewers would be valuable for clarifying remaining questions and further strengthening our work.
>
> We would be grateful if you could provide feedback on our responses, as this would allow us to better understand and address any outstanding concerns before the discussion period concludes.
>
> We remain fully committed to incorporating constructive feedback to improve our paper.
>
> Thank you for your attention and support.
>
> Best regards,
>
> Authors of Paper 5060

---

### Official Review · Reviewer_zRJ7 · 2025-10-31

**Soundness:** 2
**Presentation:** 3
**Contribution:** 3
**Rating:** 6
**Confidence:** 3

**Summary:**

This paper introduces AGDC, an autoregressive diffusion model that jointly models discrete and continuous values for variable-length sequences. It then presents ContLayNet, a large-scale benchmark comprising 334K high-precision semiconductor layout samples. Experiments on semiconductor layouts, graphic layouts, and SVGs demonstrate AGDC's superior generation performance.

**Strengths:**

**S1.** The proposed methodology is conceptually sound and neat.

**S2.** The paper introduces ContLayNet, a real-world large-scale dataset of semiconductor layout samples.

**S3.** Empirical studies (quantitative and qualitative) demonstrate the effectiveness of the proposed approach, with ablation studies provided for various design choices.

**S4.** The paper is overall easy to follow.

**Weaknesses:**

**W1.** The baselines are limited. Even though some generative models were not designed towards layout generation in particular, they may still be adaptable to this scenario.

**W2.** The paper does not discuss existing papers that combine autoregressive models and diffusion models. E.g.,

- Chen et al. Diffusion Forcing: Next-token Prediction Meets Full-Sequence Diffusion. NeurIPS 2024.

- Zhao et al. Pard: Permutation-Invariant Autoregressive Diffusion for Graph Generation. NeurIPS 2024.

- Li et al. LayerDAG: A Layerwise Autoregressive Diffusion Model for Directed Acyclic Graph Generation. ICLR 2025.

**W3.** From Figure 2, different atomic units can have continuous values of different lengths. Meanwhile, equation 7 assumes each atomic unit have exactly a single discrete value and fixed-length continuous values. This seems to suggest a limitation of AGDC in flexibility and generalizability.

**Questions:**

N.A.

---

> ### Author Response · Authors · 2025-11-18
> **Response to Reviewer zRJ7**
>
> We appreciate the reviewer’s appreciation about the qualified presentation of our paper and acknowledgment of our introduced ContLayNet. We sincerely thank the reviewers for their insightful comments and constructive suggestions, which have significantly improved the quality of our paper. We address each concern below.
>
> **[W1] Limited Baselines**
>
> Following your suggestion, we adapted TabDiff [R11], a tabular data generation method that also addresses hybrid discrete-continuous data types, to our GDS layout generation task. While we successfully trained the model on our dataset, the original architecture only supports unconditional generation and cannot handle the conditional completion task that is central to our evaluation. Consequently, we report TabDiff results only for unconditional generation in Table R2, where TabDiff underperforms compared to AGDC and other baselines.
>
> **Table R2**. Quantitative results on ContLayNet unconditional generation task, including TabDiff.
> | Methods | CLC $\downarrow$ | PDC $\downarrow$ | HSC $\downarrow$ | VSC $\downarrow$ |
> |---------|------------------|------------------|------------------|------------------|
> | TabDiff | 0.197            | 0.580            | 0.097            | 0.084            |
> | LT      | 0.230            | 0.143            | 0.038            | 0.035            |
> | DLT     | 0.248            | 0.582            | 0.060            | 0.051            |
> | AGDC    | **0.095**        | **0.123**        | **0.031**        | **0.028**        |
>
> **[W2] Discussion of existing papers**
>
> Thank you for pointing out the missing discussion of papers combining autoregressive and diffusion models. We acknowledge that [R12–14] represent important work in this area and revised the manuscript to include them in our related work.
>
> However, these methods address fundamentally different problem settings:
> - Pard [R13] and LayerDAG [R14] focus on graph generation with discrete-diffusion, specialized for node and edge prediction in graph structures.
> - Diffusion Forcing [R12] employs chunk-based denoising for continuous-valued sequences (video, trajectories) but does not fully utilize autoregressive dynamics for variable-length generation control—the model diffuses fixed-length chunks rather than autoregressively determining when to terminate generation.
>
> Our AGDC differs by addressing hybrid discrete-continuous vector generation with variable lengths, where (1) both discrete (component types, layers) and continuous (coordinates, dimensions) variables must be jointly modeled, and (2) length control is implemented to accurately determine when to stop generation (essential for completion tasks).
>
> We have uploaded the updated manuscripts with changes highlighted in blue.
>
>
> **[W3] Fixed-Length Continuous Values**
>
> While Equation 7 presents the framework with a single discrete identifier and fixed-length continuous vectors for clarity, AGDC can be extended to handle more complex scenarios.
>
> For atomic units requiring variable-length continuous values, we adopt a standardized representation where all atomic units share the same continuous vector dimensionality, but different types use only the relevant dimensions. As demonstrated in our SVG experiments and described in Appendix C.1 (lines 837-842), we represent each SVG atomic unit with a standardized 8-dimensional continuous vector (four coordinate pairs). Different command types (M, L, C) use only their relevant coordinates and ignore unused dimensions — for instance, M and L commands use 2 coordinate pairs while C commands use all 4. This unified representation enables consistent tensor operations across all command types while naturally handling varying requirements.
>
> Regarding multiple discrete attributes per atomic unit, there are two straightforward approaches: (1) merging multiple discrete attributes into a single combined class (*e.g.*, if one attribute has $K_1$ classes and the other has $K_2$ classes, create $K_1 \times K_2$ combined classes), or (2) extending the architecture with multiple classification heads, each predicting a different discrete attribute independently.
>
> [R11] Juntong Shi et al. TabDiff: a Mixed-type Diffusion Model for Tabular Data Generation. In ICLR, 2025.
>
> [R12] Boyuan Chen et al. Diffusion Forcing: Next-token Prediction Meets Full-Sequence Diffusion. In NeurIPS, 2024.
>
> [R13] Lingxiao Zhao et al. Pard: Permutation-Invariant Autoregressive Diffusion for Graph Generation. In NeurIPS, 2024.
>
> [R14] Mufei Li et al. LayerDAG: A Layerwise Autoregressive Diffusion Model for Directed Acyclic Graph Generation. In ICLR, 2025.

---

> ### Author Response · Authors · 2025-11-24
> **Request for Reviewer Engagement - Paper 5060**
>
> Dear Reviewer zRJ7,
>
> Thank you for coordinating the review process for ICLR 2026.
>
> We have submitted a comprehensive rebuttal addressing all of your concerns, but unfortunately have not yet received responses. We believe that engaging in dialogue with the reviewers would be valuable for clarifying remaining questions and further strengthening our work.
>
> We would be grateful if you could provide feedback on our responses, as this would allow us to better understand and address any outstanding concerns before the discussion period concludes.
>
> We remain fully committed to incorporating constructive feedback to improve our paper.
>
> Thank you for your attention and support.
>
> Best regards,
>
> Authors of Paper 5060

---

> > ### Comment · Reviewer_zRJ7 · 2025-11-24
> >
> > Sorry for the late response.
> >
> > Thank you for the detailed reply. I don't have further concerns or questions.
> >
> > For better presentation, I suggest you put the answer to W3 in an appendix section and add a reference to it after equation 7.
> >
> > I think my previous evaluation is fair, so I will keep it.

---

> > > ### Author Response · Authors · 2025-11-25
> > >
> > > Thank you so much for consistently positive comments. We are very happy to hear that our response clarified your concerns.
> > >
> > > Authors of papet 5060

---

### Official Review · Reviewer_65Z6 · 2025-11-01

**Soundness:** 2
**Presentation:** 1
**Contribution:** 2
**Rating:** 2
**Confidence:** 2

**Summary:**

The paper presents AGDC, a novel autoregressive framework that jointly models discrete and continuous values within variable-length sequences. It predicts discrete values through categorical prediction and continuous values using diffusion-based probabilistic models. The paper presents experiments across semiconductor layouts (with a new large-scale benchmark), graphic layouts, and text-to-SVG synthesis. The results shows that AGDC outperforms existing discretization-based and fixed-schema methods.

**Strengths:**

* Joint models of discrete and continuous values in autoregressive model is useful for modeling and generating complex data.
* The paper proposes a new benchmark for circuit layout.
* The experiments are showing that the encoding and generation work on multiple problems, indicating the generality of the proposed approach. Out of the three parts of the experiment, the circuit layout is the most interesting case study.

**Weaknesses:**

* The paper is light on theoretical contributions. The technical approach seems as a combination of known models, which translates to explaining “how” the approach works, leaving “why” out.
* The motivation behind EOS logit adjustment was not clear.
* The results on the chip layout problem seem  notable. The table shows improvement in several existing metrics in the domain. However, these metrics seem very problem specific and their motivation is not well explained for the general audience. The qualitative notion of “more balanced and well-distributed layers” is hard to justify by representative images only; a quantitative analysis is likely necessary.
* Additionally, it is difficult to grasp the evaluation setup and metrics for this study without going to the appendix. The evaluation metrics for circuit layout generation should be described in more detail in the main body of the paper, by moving some text from the appendix.
* The applications on other two case studies in the evaluation (graphics layout and text-to-svg) seem less impactful.

**Questions:**

No questions. My overall suggestion is that the authors revise the paper to emphasize the merits of their algorithmic contribution and provide more background for the readers who are not circuit design experts.

---

> ### Author Response · Authors · 2025-11-18
> **Response to Reviewer 65Z6 (Part 1)**
>
> Thank you for recognizing the utility of our joint discrete-continuous modeling approach and the contribution of our benchmark. We address your concerns below.
>
> **[W1] Light theoretical contributions**
>
> We would like to clarify that the primary contribution of AGDC is a principled solution to a long-standing technical challenge, rather than the introduction of new theoretical frameworks.
>
> **Problem Motivation.**
> Hybrid discrete-continuous sequence generation presents fundamental obstacles that existing approaches fail to resolve. Discretization-based methods exhibit exponential vocabulary growth as precision increases (Section 3.1, Table B), making them impractical for high-precision tasks. Conversely, non-autoregressive models such as DLT cannot reliably control sequence length, resulting in unstable outputs on variable-length datasets (Tables 2–3, Figure B). AGDC directly addresses this gap by preserving continuous precision while retaining the controllability and expressiveness of autoregressive modeling.
>
> **Technical Novelty.**
> While AGDC leverages established components (*e.g.*, transformers and diffusion models), the novelty lies in their integration and in the mechanisms required to make them operate coherently in a single autoregressive framework. Specifically:
> - The unified latent-conditioning mechanism enables discrete categorical prediction and continuous diffusion sampling to be jointly performed within each autoregressive step.
> - The MLP-based EOS logit adjustment resolves context-dependent termination in hybrid output spaces, a challenge not handled by existing autoregressive or diffusion models.
> - The differentiable length-regularization objective provides end-to-end optimization of sequence length—crucial for variable-length hybrid sequences.
>
> These contributions form a coherent framework that neither discrete autoregressive models nor diffusion models can achieve alone.
>
> **Empirical Validation.**
> Ablation studies (Tables 2–4, Figure 6) show that each component materially contributes to performance. Moreover, consistent gains across three distinct domains demonstrate that AGDC captures general principles rather than task-specific heuristics. We believe that solving a previously unaddressed modeling problem with a principled and generalizable framework constitutes a meaningful contribution to the machine learning community.
>
> **[W2] Motivation behind EOS logit adjustment**
>
> In variable-length generation, the model must determine when to terminate sequences based on context. Standard categorical prediction treats EOS as just another token, making termination decisions solely through the softmax distribution. In practice, we found that this often leads to overly long sequences or cases where EOS is never produced, causing the model to default to the maximum length.
>
> The EOS logit adjustment mechanism mitigates this issue by using an MLP to incorporate sequence-level context from the latent representation $\mathbf{z}^i$ when predicting termination. This provides an auxiliary, context-aware signal that improves the model’s ability to identify correct stopping points based on accumulated history. As shown in our ablation studies (Tables 2–3), removing this mechanism consistently degrades performance across all metrics.

---

> ### Author Response · Authors · 2025-11-18
> **Response to Reviewer 65Z6 (Part 2)**
>
> **[W3] Concerns for metrics**
>
> We would like to clarify the motivation behind our evaluation metrics and provide additional quantitative analysis in visualized images as suggested.
>
> **Motivations of Metrics.**
> As described in Section 5.2 of the updated paper, the metrics we employ directly measure the functional correctness and geometric precision of semiconductor circuit layouts. Unlike traditional image-based metrics, these metrics capture underlying logic and physical constraints that determine the overall performance and functionality of the layout. This is why domain-specific metrics are essential for evaluating layout generation quality.
>
> **Visual Metric Analysis (FID).**
> While acknowledging that image-based metrics cannot fully capture the physical and logical properties of circuit layouts, we have conducted FID analysis to complement our domain-specific evaluation. Compared with image-based metrics, **AGDC still achieves the best FID scores** among all methods. AGDC maintains superior performance, demonstrating better visual quality alongside functional correctness. The results are summarized in Table R3. Note that the FID score between our validation set and test set was 46.31.
>
> **Table R3.** Image FID score evaluation results.
> |                               | AGDC - Test set | LT - Test set | DLT - Test set |
> |-------------------------------|-----------------|---------------|----------------|
> | Completion (given 50 layers)  | **56.10**       | 87.80         | 96.08          |
> | Completion (given 100 layers) | **43.79**       | 47.90         | 77.25          |
>
> The FID analysis reveals that AGDC generates not only functionally more correct but also visually cleaner layouts with fewer artifacts, further validating the qualitative improvements mentioned in our paper.
>
> **[W4] Details of evaluation metrics should be moved to the main body**
>
> We apologize for the inconvenience. We have moved the detailed explanation about the ContLayNet benchmark to the main body of the paper. You may check the revised version of the newly uploaded manuscript.

---

> ### Author Response · Authors · 2025-11-18
> **Response to Reviewer 65Z6 (Part 3)**
>
> **[W5] Impact for graphic layout generation / Text-to-SVG applications**
>
> We acknowledge the reviewer’s observation and believe this actually highlights the value of ContLayNet as a benchmark for the machine learning community. Most existing benchmarks (*e.g.*, images) lack clear metrics where precision directly impacts functional correctness, making it difficult to rigorously evaluate methods that preserve continuous values.
>
> The graphic layout generation and text-to-SVG experiments serve important complementary purposes. While precision requirements may be less critical in these domains than in semiconductor design, we strongly believe that the fundamental reason for using continuous vector representations is to avoid sacrificing precision through discretization. This is particularly important for SVG (Scalable Vector Graphics), where the “scalable” property inherently requires infinite precision to enable lossless scaling at arbitrary resolutions.
>
> Graphic layout generation is a well-established research area in the machine learning community with extensive prior work [R1–10]. Our results demonstrate that AGDC achieves competitive performance while operating in the native continuous space without discretization, and demonstrates superior performance compared to the discretization-based method LT in high-precision settings (Table 3, Figure 7). For text-to-SVG, our work is the first to preserve continuous SVG coordinates without discretization, and the failure of IconShop at higher precision levels validates our core thesis about the scalability limitations of discretization-based approaches (Table 4, Figure 7).
>
> Together, these experiments demonstrate that our framework is effective and generalizable across diverse application domains, supporting our contribution to the broader machine learning community.
>
> [R1] Peirong Zhang et al. Smaller But Better: Unifying Layout Generation with Smaller Large Language Models. In IJCV, 2025.
>
> [R2] Jian Chen et al. Towards aligned layout generation via diffusion model with aesthetic constraints. In ICLR, 2024.
>
> [R3] Zecheng Tang et al. LayoutNUWA: Revealing the Hidden Layout Expertise of Large Language Models, In ICLR, 2024.
>
> [R4] Julian Jorge Andrade Guerreiro et al. Layoutflow: Flow matching for layout generation. In ECCV, 2024.
>
> [R5] Yilin Wang et al. Dolfin: Diffusion Layout Transformers without Autoencoder. In ECCV, 2024.
>
> [R5] Elad Levi et al. DLT: Conditioned layout generation with Joint Discrete-Continuous Diffusion Layout Transformer. In ICCV, 2023.
>
> [R6] Junyi Zhang et al. LayoutDiffusion: Improving Graphic Layout Generation by Discrete Diffusion Probabilistic Models. In ICCV, 2023.
>
> [R7] Jiawei Lin et al. LayoutPrompter: Awaken the Design Ability of Large Language Models. In NeurIPS, 2023.
>
> [R8] Chin-Yi Cheng et al. PLay: Parametrically Conditioned Layout Generation using Latent Diffusion. In ICML, 2023.
>
> [R9] Kamal Gupta et al. LayoutTransformer: Layout Generation and Completion with Self-attention. In ICCV, 2021.
>
> [R10] Jianan Li et al. LayoutGAN: Generating Graphic Layouts with Wireframe Discriminators. In ICLR, 2019.

---

> > ### Comment · Reviewer_65Z6 · 2025-11-28
> >
> > Thank you for the response. The explanations offered in W2-W4 should be included in the main paper. My concerns about W1 and W5 remain, as well as the general impression that the paper attempts to cover too much ground. As a result, it is not being able to highlight the key technical contributions for the broad ML community and why it works. I will thus keep the current rating.

---

> ### Author Response · Authors · 2025-11-24
> **Request for Reviewer Engagement - Paper 5060**
>
> Dear Reviewer 65Z6,
>
> Thank you for coordinating the review process for ICLR 2026.
>
> We have submitted a comprehensive rebuttal addressing all of your concerns, but unfortunately have not yet received responses. We believe that engaging in dialogue with the reviewers would be valuable for clarifying remaining questions and further strengthening our work.
>
> We would be grateful if you could provide feedback on our responses, as this would allow us to better understand and address any outstanding concerns before the discussion period concludes.
>
> We remain fully committed to incorporating constructive feedback to improve our paper.
>
> Thank you for your attention and support.
>
> Best regards,
>
> Authors of Paper 5060

---

> ### Author Response · Authors · 2025-11-28
>
> Dear Reviewer 65Z6,
>
> Thank you for the time you have invested in reviewing our submission.
>
> We would like to kindly ask for clarification regarding your comment. In the rebuttal, we elaborated on the theoretical contribution---namely, the advantages of jointly modeling discrete and continuous elements within a unified autoregressive framework---and demonstrated that our approach consistently outperforms prior work on well-studied tasks such as SVG generation and graphic layout modeling.
>
> Nonetheless, your final statement that “the paper attempts to cover too much ground” does not include specific details, which makes it difficult for us to understand the technical limitations you are referring to. If you could indicate which aspects you believe remain insufficient, we would be more than willing to provide further clarification.
>
> We sincerely appreciate your feedback and would be grateful for any more concrete comments you can offer so that we can better address your concerns.
>
> Best regards,
>
> Authors of Paper 5060

---

### Author Response · Authors · 2025-12-02
**Summary for Area Chairs**

Our paper proposes *AGDC*, a unified autoregressive framework that jointly models discrete and continuous values within variable-length sequences, addressing a fundamental limitation: discretization-based methods exhibit exponential vocabulary growth and performance degradation at high precision (Table A), while non-autoregressive approaches struggle with controlling sequence length. The framework introduces three key technical contributions: (1) unified latent-conditioning mechanism enabling joint prediction, (2) MLP-based EOS logit adjustment, (3) differentiable length-regularization objective, validated through ablation studies (Tables 2–4, Figure 6). We additionally present *ContLayNet*, a benchmark comprising 334K high-precision semiconductor layout samples with specialized metrics measuring functional correctness, providing the first rigorous testbed for high-precision generative modeling.

We have **comprehensively addressed all reviewer concerns** through new experiments and manuscript revisions, including: FID analysis validating visual quality, TabDiff baseline comparisons, and extensive inference timing analysis demonstrating competitive speed with superior quality. Due to changes in the review process, we provide this summary to assist evaluation.

## Reviewer 65Z6 (Rating: 2 / Confidence: 2)
**Main concerns**: Light theoretical contribution; unclear EOS logit adjustment motivation; metrics need explanation; limited impact of graphic layout generation/SVG experiments.

**Our response**:
- Clarified AGDC’s primary contribution is a principled solution to a long-standing technical challenge rather than new theoretical frameworks.
- Added FID analysis (Table R3) showing superior visual quality alongside functional correctness measured by DRC metrics.
- Moved DRC metric details to the main body as requested.
- Explained graphic layout generation and SVG applications serve important complementary purposes: graphic layout generation is a well-established research area with extensive prior work, while SVGs' “scalable” property fundamentally requires continuous precision for lossless scaling.

**Follow-up**: The reviewer acknowledged W2–W4 were addressed. For W1 and W5, the reviewer indicated concerns remain, though the review lockdown prevented further technical discussion or clarification. We have provided detailed responses with additional experiments and analysis for these points, and **believe they substantively address the raised concerns.** We remain available for any further clarification that would be helpful, and we welcome the Area Chair's evaluation of our technical responses to these points.

## Reviewer zRJ7 (Rating: 6 / Confidence: 3)
**Main concerns**: Limited baselines; missing related work; fixed-length continuous-value assumption.

**Our response**:
- Added TabDiff baseline showing AGDC outperforms across all metrics (Table R2).
- Expanded related work including all the related works that the reviewer has mentioned, clarifying these address fundamentally different settings.
- Clarified Equation 7 is simplified for presentation; AGDC extends to variable-length scenarios through standardized representations (Appendix C.1).

**Follow-up**: The reviewer stated they had no further concerns or questions.

## Reviewer r7X6 (Rating: 4 / Confidence: 2)
**Main concerns**: Inference speed; unclear contribution to the ML community.

**Our response**:
- Comprehensive timing analysis (Table R1): AGDC with 20 diffusion steps achieves **15% faster inference** than optimized LayoutTransformer (617s vs. 766s) **while significantly outperforming DRC metrics.**
- Clarified three-fold contributions to the ML community: (1) Methodological contribution with a unified framework and key technical contributions, (2) ContLayNet benchmark enabling rigorous evaluation where precision impacts functional correctness, (3) broader applicability to other domains such as molecular design, music generation, and CAD modeling.

**Follow-up**: **We provided comprehensive technical responses** addressing the reviewer's questions. The review lockdown prevented further discussion, but we remain available for any clarification that would be helpful regarding our responses to their concerns.

## Summary
We provided detailed technical responses with substantial new experiments addressing all concerns. Reviewer zRJ7 explicitly confirmed satisfaction. Despite our comprehensive rebuttal, Reviewers r7X6 and 65Z6 could not engage further due to the review lockdown. We believe this work addresses important challenges in high-precision generative modeling for structured vector representations, with direct applications to semiconductor circuit design where precision requirements impact functional correctness. Given the detailed responses, we respectfully request the Area Chairs consider our rebuttal carefully, particularly from (high-precision) generative modeling perspectives, when forming the recommendation.

---

### Meta-Review · Area_Chair_Wxga · 2026-01-07

**Summary:**

This paper introduces AGDC, a unified autoregressive framework that jointly predicts discrete tokens and high-precision continuous values within variable-length sequences. However, the submission raised significant concerns. Reviewers questioned (i) limited theoretical novelty, (ii) unclear motivation for the EOS-logit-adjustment module, (iii) insufficient baseline diversity, (iv) possible rigidity of the fixed-length continuous representation. Although the rebuttal clarified some aspects, core concerns about more persuasive motivation of the proposed AGDC and the contribution of this work to the machine learning community remained unresolved in the Introduction section. Based on the above considerations, I recommend rejection, while encouraging the authors to further develop this promising direction.

**Reviewer Concerns:**

Addressed: baseline shortage, metric explanations & FID evidence, inference-speed data.

Still outstanding: EOS-logit motivation, depth of theoretical/novelty claim, contribution to the broader ML community, rigidity of fixed-length continuous vector.

**Reviewer Scores:**

Reviewer zRJ7 would have kept the positive score. Reviewer 65Z6 and Reviewer r7X6 would have kept their negative scores.

---

### Decision · Program_Chairs · 2026-01-26

Reject